# Intraoperative Neuromonitoring and Optical Magnification in the Prevention of Recurrent Laryngeal Nerve Injuries during Total Thyroidectomy

**DOI:** 10.3390/medicina58111560

**Published:** 2022-10-30

**Authors:** Menelaos Karpathiotakis, Valerio D’Orazi, Andrea Ortensi, Andrea Biancucci, Rossella Melcarne, Maria Carola Borcea, Chiara Scorziello, Francesco Tartaglia

**Affiliations:** 1Department of Surgical Sciences, Sapienza University of Rome, Viale Regina Elena 324, 00161 Rome, Italy; 2Division of General Surgery, Fabia Mater Hospital, Via Olevano Romano 25, 00171 Rome, Italy

**Keywords:** total thyroidectomy, recurrent laryngeal nerve paralysis, intraoperative neuromonitoring, optical magnification

## Abstract

*Background and Objectives*: Recurrent laryngeal nerve (RLN) paralysis is a fearful complication during thyroidectomy. Intraoperative neuromonitoring (IONM) and optical magnification (OM) facilitate RLN identification and dissection. The purpose of our study was to evaluate the influence of the two techniques on the incidence of RLN paralysis and determine correlations regarding common outcomes in thyroid surgery. *Materials and Methods*: Two equally sized groups of 50 patients who underwent total thyroidectomies were examined. In the first group (OM), only surgical binocular loupes (2.5×–4.5×) were used during surgery, while in the second group (IONM), the intermittent NIM was applied. *Results*: Both the operative time and the length of hospitalization were shorter in the OM group than in the IONM group (median 80 versus 100 min and median 2 versus 4 days, respectively) (*p* < 0.05). The male patients were found to have a five-fold higher risk of developing transient dysphonia than the females (adjusted OR 5.19, 95% IC 0.99–27.18, *p* = 0.05). The OM group reported a four-fold higher risk of developing transient hypocalcemia than the IONM group (OR 3.78, adjusted OR 4.11, *p* = 0.01). Despite two cases of temporary bilateral RLN paralysis in the IONM group versus none in the OM group, no statistically significant difference was found (*p* > 0.05). No permanent RLN paralysis or hypoparathyroidism have been reported. *Conclusions*: Despite some limitations, our study is the first to compare the use of IONM with OM alone in the prevention of RLN injuries. The risk of recurrent complications remains comparable and both techniques can be considered valid instruments, especially if applied simultaneously by surgeons.

## 1. Introduction

Thyroid surgery is a common procedure; although it is performed with a standardized technique, there is still the risk of severe complications. Undoubtedly, recurrent laryngeal nerve (RLN) paralysis is a life-threatening condition that, even when unilateral and transient, can significantly affect the quality of life. Early identification and visualization of RLN during thyroidectomy is now considered the gold standard method to prevent injuries [1]. Nevertheless, the RLN may not be visualized in up to 11.5% of thyroidectomies, and this rate increases to 66% in patients with recurrent diseases [2].

Furthermore, the direct visualization of RLN reveals only 10% to 14% of injuries [3] and the anatomical integrity of RLN does not always relate to a normal vocal cord (VC) function postoperatively [4]. Thus, different technological innovations have been introduced in thyroid surgery in the last decades. A microsurgical approach with optical magnification (OM) systems is stated to increase precision during thyroidectomy; noble anatomical structures, such as nerves, could be identified with major accuracy [5]. Moreover, to obtain information on RLN functionality, intraoperative neuromonitoring (IONM) was introduced.

The purpose of our study was to evaluate the influences of these two techniques on the incidence of RLN paralysis and determine correlations regarding common outcomes in thyroid surgery.

## 2. Materials and Methods

From October 2018 to February 2020, the data collected from two equally sized groups of 50 patients who underwent total thyroidectomy were prospectively analyzed. For RLN identification and dissection, in the first group, optical magnification only (OM group) was used, while in the second group, intraoperative neuromonitoring was applied (IONM group). In the IONM group, optical magnification was not used.

Exclusion criteria were previous thyroid surgery, lobectomy, neck irradiation, concomitant parathyroidectomy, lymphadenectomy, and minimally invasive procedures (MIVAT, TOETVA). Written informed consent was provided by all patients and the study was approved by the Local Bioethics Committees of Policlinico “Umberto I” and “Fabia Mater” Hospital.

Total thyroidectomy with a standard technique was performed by experienced endocrine surgeons who were less familiar with these new technologies. Hemostasis was obtained using bipolar electrocautery and ligatures.

The surgical devices selected in the first group were binocular loupes with 2.5×–4.5× magnification and a focal range of about 17 inches, combined with a head-mounted led coaxial light to increase operative field brightness. In the second group, the NIM 3.0 by Medtronic (Florida, FL, USA) was applied in the intermittent modality only for the IONM. An endotracheal tube (Flex Medtronic, 7–8 mm of diameter) with surface electrodes incorporated, was used for the electromyographic signal acquisition. To ensure precise localization of the electrodes between the vocal cords, the tube was positioned 20–22 cm from the superior dental arch. Conventionally, a loss of signal (LOS) was defined as a decrease in the RLN amplitude of fewer than 100 MicroV or a complete loss of amplitude after proper vagus stimulation on the upper threshold limit (1–2 mA). Typically, the LOS was distinct into two types: Type 1, in which a defined level of LOS was detected corresponding to a segmental RLN lesion; Type 2, in which a total LOS was detected regarding both RLN and vagus nerve [6,7].

All patients had their normal vocal cord functions verified by a ‘fibrolaringoscopy’ preoperatively. We collected data relating to the clinical assessment, complications during hospitalization, and post-operative dysfunctions. The parameters recorded were: sex, age, admission diagnosis, thyroid Doppler ultrasonography, hyperfunction, thyroiditis, preoperative cytology, duration of the operation, RLN injuries, hemorrhage, hypocalcemia (transient or permanent), perioperative cortisone therapy, dysphonia, length of hospitalization, and histology report.

The follow-up period was 6 months and each patient underwent a ‘fibrolaringoscopy’ and calcium serum level dosage. We did not record any cases of mortality in our series.

Statistical analysis was performed using SAS software, version 6.1. Continuous variables are expressed as medians (interquartile range, IQR), while categorical variables are expressed as proportions and percentages. Student’s t-test was used to compare continuous variables with normal distribution. The chi-square and Fisher’s exact tests were used to evaluate differences between categorical variables, toward the methodology of the regression logistic model and backward analysis, verifying the accuracy of the outcomes and estimating the odds ratio (OR). The level of significance was set at *p* < 0.05.

This is a prospective study where the patients were casually and alternatively assigned to one of the two groups (OM or IONM), based on the progressive booking number to undergo total thyroidectomy in our institution, with up to 50 patients in each group.

## 3. Results

The two groups were homogeneous in age (median OM 53.5 years; IONM 55 years), sex (Female OM 82%; IONM 84%), diagnosis of admission (multinodular goiter OM 96%; IONM 94%), and follow-up outcomes. No statistically significant differences in the presence of hyperfunction, thyroiditis, and preoperative cytology were found, and the distribution of the nerves-at-risk (malignancy, multinodular cervicomediastinal goiter, hyperthyroidism) was similar in the two groups. (Table 1).

The operative time in the OM group (median 80 min, IQR 70–90 min) was shorter than that in the IONM group (median 100, IQR 90–119 min) (*p* < 0.05). The median hospital stay in the OM group was 2 days (IQR 2–2) while in the IONM group was 4 days (IQR 4–5). The Box and Whisker plots in Figure 1 show the distributions of these variables.

Postoperative transient hypocalcemia (total serum calcium <8 mg/dl) was recorded in 17 patients (34%) in the OM group and in 6 patients (12%) in the IONM group. Both univariate and multivariate analyses reported statistically significant differences regarding transient hypoparathyroidism in the two groups, as represented by the ROC curve in Figure 2, with the OM group having a four-fold higher risk of developing transient hypocalcemia than the IONM group (OR 3.78, adjusted OR 4.11, *p* = 0.01) (Table 2).

Only five patients in the OM group and one patient in the IONM group required oral or intravenous calcium treatments for symptomatic hypocalcemia and all patients had normal calcium levels within a week of surgery. No cases of permanent hypoparathyroidism were reported. Six patients (12%) in the IONM group and five patients (10%) in the OM group experienced transient postoperative dysphonia, which regressed 5–7 days after surgery. The multivariate logistic regression method adjusted by group and gender, documented for this outcome a statistically relevant difference (ROC curve-Figure 2). Thus, the males in our study had a five-fold higher risk of developing transient dysphonia than the females (adjusted OR 5.19, 95% IC 0.99–27.18, *p* = 0.05) (Table 2).

No cases of unilateral or bilateral RLN paralysis were recorded in the OM group. Contrarily, two patients in the IONM group (4%) had temporary bilateral RLN paralysis documented postoperatively. However, no statistically significant difference was found between the two groups (*p* > 0.05).

Cortisone therapy with 8 mg of intravenous dexamethasone was administered intraoperatively in each patient, according to the anesthetic protocol of our Institute. However, the needs for postoperative intravenous cortisone administration were significantly higher in the IONM group (16%) than in the OM group (2%) (*p* < 0.05).

The histological examination revealed a total of 20 (40%) patients with thyroid cancer in the OM group and 15 (30%) in the IONM group; the presence of overall carcinomas in the OM group was statistically significant (*p* < 0.05). The follow-up at 6 months reported all patients with preserved vocal cord motility and no cases of permanent RLN paralysis or hypoparathyroidism were detected.

## 4. Discussion

During total thyroidectomy, the risk of permanent RLN paralysis, even in experienced centers, is estimated to be nearly 1% for multinodular goiters [8], while in the case of reoperations, it can be quite high (30%) [9]. A recent systematic review reported an incidence of temporary and permanent RLN palsy of 9.8% and 2.3%, respectively [10]. Additionally, data from a wide national database documented a two-fold increase in RLN paralysis rates in patients who routinely undergo postoperative laryngoscopy [3]. Therefore, the incidence of RLN palsy is often underestimated and there are still RLN complications in a total thyroidectomy.

Regarding the need for early RLN identification, Jatzko et al. conducted a multicenter study, which was fundamental in dissolving controversies; the incidences of temporary and permanent RLN paralyses in the group without RLN identification were much higher than in the group with nerve identification (7.9% and 5.2% vs. 2.7% and 1.2%, respectively) [11].

As reported by Dionigi, the four most common mechanisms of RLN injury during thyroidectomy, in order of frequency, are: traction, thermal stress, compression, and clamping [12]. Traction occurs mainly during the dissection of Berry’s suspensory ligament [13,14] and 80% of all RLN injuries associated with vocal cord palsy are due to stretching [12].

Undoubtedly, OM systems have important roles in preventing most causes of RLN injuries. As suggested by Testini et al., using 2.5× magnification loupes is extremely useful for RLN identification, especially in high-risk thyroidectomies, significantly reducing the operative times [15]. However, for Pata et al., the use of OM had no effect on the rate of RLN injuries [16]. Interestingly, the literature offers only a few studies on the use of OM during thyroidectomies.

In our series, we used OM loupes during the entire operation, experiencing great comfort and total freedom in moving the upper body [5]. We agree that OM systems can be considered helpful in thyroid surgery, by increasing the definition of the operating field and growing the surgeon’s visual acuity [5,17]. Therefore, RLN dissection can be more precise and safer, minimizing the risk of traction or thermal-related injuries (Figure 3 and Figure 4).

The number of thyroidectomies performed with IONM is steadily increasing, reflecting the rapid popularization of this method [18]. The application of IONM is subordinate to the equipment availability, age, training, and surgeon’s experience [19,20].

Worldwide the use of IONM is widespread, with some statistical differences among countries. For instance, in Germany, more than 90% of thyroidectomies are performed using IONM; in the United States, it is used in up to 40% of operations. In France, the rate is nearly 15% of thyroid procedures [6,18]. In Italy, from 2007 to 2013, the number of thyroidectomies performed with IONM steadily increased from 253 to 5100, respectively, reflecting the rapid popularization of this method [18].

On the other hand, magnification techniques (surgical loupes or surgical microscopes) are not yet standardized in total thyroidectomies. Few data are available in the literature, as reported by a recent meta-analysis [17], with different methods and degrees of magnification applied in heterogeneous studies, of which, only three were RCTs. Therefore, more prospective trials are necessary to better define the impacts of magnification techniques.

There is evidence in the literature that IONM facilitates the early identification of RLN lesions, even in those nerves that can appear visually and anatomically intact [21]. In addition, IONM permits us to understand the mechanisms of injuries and, therefore, intraoperatively predict the nerve functions; an injury that occurs on a morphologically intact RLN regularly results in temporary reversible vocal cord paralysis rather than permanent damage [22,23]. The application of IONM is also associated with the possibility of changing surgical planning intraoperatively, in accordance with the concept of “stage thyroidectomy”; in case of a LOS during the first lobe dissection, the contralateral lobectomy should be postponed, avoiding the risk of bilateral RLN paralysis [4,13,21]. Furthermore, IONM can identify atypical anatomical nerve variations, which are linked with high risks of lesions [24]. Similarly, as reported by two meta-analyses, the use of IONM significantly decreases the temporary or permanent vocal cord palsy ratio [25,26]; these results do not depend on the surgeon’s experience level [27]. Finally, the velocity of RLN isolation is improved using IONM compared to visual identification alone [21,28].

Some authors stated that only continuous IONM can prevent the RLN from traction injuries [29]. However, the continuous IONM probe might register a decline during the mobilization of the thyroid gland [7]. Interestingly, up to 70% of the injured nerves in intermittent modality recovered their functions intraoperatively [6,8]. Thus, we found it reasonable not to use the IONM in the continuous modality in our series.

Despite the different high-volume studies, summarized by reviews and meta-analyses [6,25,26] that claimed the reduction of RLN injuries with IONM, opinions are not always concordant. Some authors determined no significant differences in the rates of transient or permanent RLN injuries between IONM and visual identification alone [30,31,32,33,34]. Similarly, a limited number of studies reported advantages in using IONM mostly during high-risk thyroidectomies, such as reoperation, malignancy, Grave’s disease, or cervicomediastinal goiters [21,35,36,37,38].

Furthermore, there is discordance regarding the cost–benefit ratio of using the IONM, as the equipment may amount to 7% of the hospital expenses charged for thyroid surgery [39]. However, the use of IONM has proved to be cost-effective when the rate of RLN paralysis is reduced by up to 50.4% [40].

In our study, the most relevant and surprising data were the presence of two transient bilateral RLN paralyses in the IONM group, which required acute management of the airways and the transfer of both patients to the intensive care unit until recovery. This resulted in the extensions of their hospital stays to 14 and 22 days, respectively, and clearly influenced the hospitalization lengths with a statistically significant difference in favor of the OM group (median 2 vs. 4 days, *p* < 0.05).

In both cases of temporary bilateral RLN paralysis, at the end of each thyroid lobe dissection, the RLN function was tested with the IONM and was found preserved, otherwise, a stage thyroidectomy should have been performed [21]. Only after revising the hemostasis on the trachea axis did we record a bilateral LOS of type 1. Hence, we must assume that the RLN injuries were likely caused by the transfer of thermal energy during the electrocoagulation maneuvers, and this might have affected, bilaterally, the intralaryngeal branches of the RLNs. As reported by the literature, thermal stress is one of the most significant injuries in terms of the risk of permanent VC palsy [7,12,21], and, in accordance with other authors, moderate to severe thermal injuries of the RLNs involve an increased rate of permanent VC palsy that needs a longer recovery time [41]. Regarding the other 48 patients of the IONM group, we recorded 3 cases of LOS type 2 during the dissection of the second lobe, which did not result in a postoperative RLN deficit (an event also reported in the literature). Indeed, a LOS registered intraoperatively with intermittent IONM was not associated with vocal cord palsy in up to 33% of the cases [42]. We, therefore, believe that these cases were due to the mispositioning of the surfaces of the electrodes, which likely occurred due to excessive anterior medial traction on the thyroid lobe during the dissection of Berry’s ligament [7,13]. Despite this unexpected result in our series, we must not forget that the use of IONM is not yet correlated with a zero incidence rate of RLN lesions, as evidenced by the literature. For instance, a meta-analysis of 30,992 total thyroidectomies reported that the incidence of bilateral RLN paralysis in the group of IONM was average at 2.43% versus 5.18% in the group without IONM [26].

The duration of the intervention seemed to be significantly shorter in the OM group (median 80 vs. 100 min, *p* < 0.05); this does not surprise us since it is well-known how the setup and the necessary measurements with the use of IONM might be quite dispendious. Equipment-related technical problems during the application of IONM occurred more frequently in the first 50 thyroidectomies and numerically corresponded to our group of patients. As claimed by Dionigi et al., the rate of positive outcomes of IONM rose over 90% after 50 applications and the operative time decreased significantly after 100 thyroidectomies [43].

Regarding the importance of cortisone therapy, we routinely used the administration of dexamethasone 8 mg i.v. during the thyroidectomy, despite controversies in the literature. The authors of the POLT study reported no effect of the intraoperative use of corticosteroids in reducing the rate of postoperative vocal cord paralysis after LOS [13]. Contrarily, the administration of steroids for type 2 lesions could reduce the rate of postoperative RLN palsy, and the recovery time after LOS (in cases with temporary vocal cord palsy) is lower [4].

Finally, the risk of transient postoperative hypocalcemia was found to be four times higher in the OM group than that in the IONM group (34% patients vs. 12% patients, OR 3.78, adjusted OR 4.11, *p* = 0.01). We assume that this high rate in the OM group could be explained by the histopathology reports. Although the nerves-at-risk were equally distributed between the groups, the histology revealed a statistically significant difference in thyroid cancer in the OM group (40% vs. 30%, *p* < 0.05) and these data were consistent with the results of FNA and cytology performed preoperatively on those patients. However, the presence of cancer did non modify the operative method in the OM group as a lymphadenectomy was not required (one of the exclusion criteria of our study). Thus, the intraoperative finding of nodules suspected of malignancy requires a meticulous dissection to ensure radical excision; the maneuvers performed to preserve the parathyroid glands might temporarily compromise the vascularization and, therefore, their functions. [44]. Furthermore, the histopathology reported on the specimen the avulsion of one parathyroid gland in two cases in the OM group and in five cases in the IONM group, supporting the thesis that OM is advantageous for accurate identification and dissection of both RLN and parathyroid glands [5]. Finally, no statistical difference was found regarding permanent hypoparathyroidism between the two groups and the data are in line with those of our previous experiences [5,44,45,46,47,48,49,50].

As stated in the literature [5,15,16,17], the use of magnification techniques is not yet well defined in total thyroidectomies, while the use of IONM is widespread [6,18]. We are aware that more prospective trials are necessary to validate the use of OM in thyroid surgery and a study comparing the results of OM + IONM versus OM alone is about to begin at our institution.

The only biases in our study are related to the use of IONM in intermittent modality only and to the limited number of patients enrolled where the surgeons involved just reached the learning curve for these two techniques. Despite the limitations, this is a prospective study based on data collected directly from the clinical assessment and the perioperative course of each patient and not through the data reported in the medical records. Therefore, a new study with the use of IONM in the continuous modality and with a larger cohort of patients could certainly provide us with further information.

## 5. Conclusions

To our knowledge, this is the first study in the literature that directly compares the use of IONM with OM alone in the prevention of RLN injuries during a total thyroidectomy.

While OM is advantageous for the accurate identification, isolation, and dissection of the RLN, IONM allows intraoperative assessments and mapping of the nerve’s integrity and functionality. Regarding the two cases of temporary bilateral RLN paralyses in the IONM group, no statistically significant differences were found and the overall risk of RLN lesions remained comparable in the two groups.

Our results show that both techniques can be considered as standards or complementary methods in the future, at least in highly specialized centers. Undoubtedly, by associating IONM and OM with microsurgery techniques, as well as adequate training, we expect to find better outcomes, as they provide greater confidence to surgeons, even if less experienced; multicenter randomized trials are necessary to validate the results. The higher costs and the need for specific equipment would be repaid by the decrease in morbidity and lower hospitalizations.

## Figures and Tables

**Figure 1 medicina-58-01560-f001:**
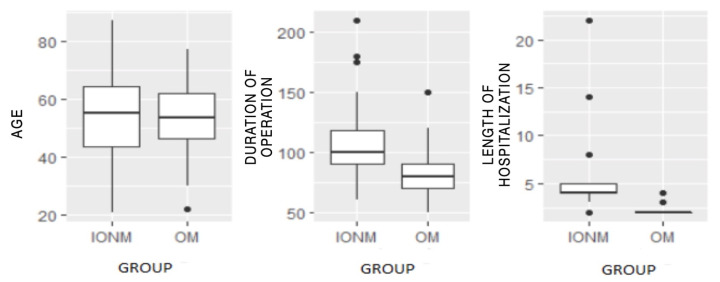
Box and Whisker plots of continuous variables; age, duration of the operation, and length of hospitalization. There were no statistical differences in the median age between the two groups, while for the other variables, there were evident differences.

**Figure 2 medicina-58-01560-f002:**
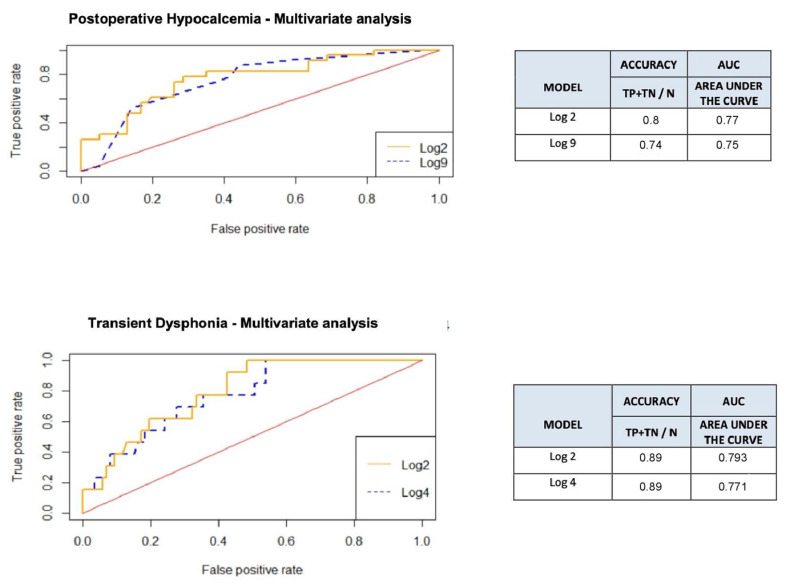
ROC curves and accuracy of the multivariate analysis of the transient postoperative hypocalcemia and transient dysphonia outcomes. The most appropriate logistic models were extrapolated through backward analysis, and compared with those models including more or fewer variables but with similar statistical significances. For both outcomes, the Log2 model reported a higher accuracy and was chosen for the analysis.

**Figure 3 medicina-58-01560-f003:**
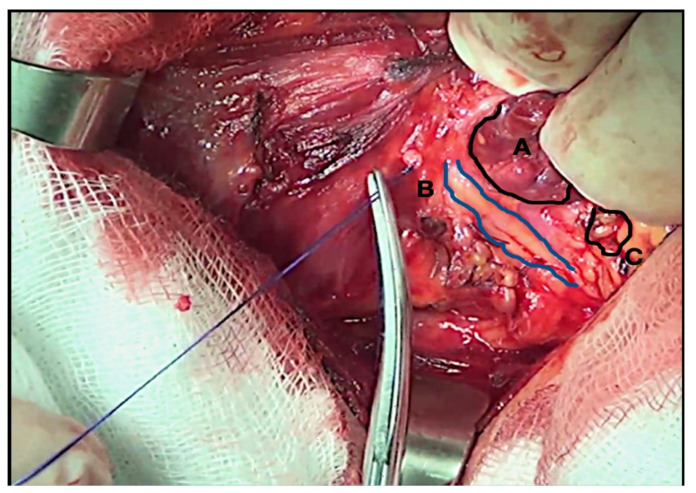
RLN dissection with optical magnification 2.5×. A Thyroid gland. B Right-sided RLN identified and isolated. C Inferior parathyroid gland.

**Figure 4 medicina-58-01560-f004:**
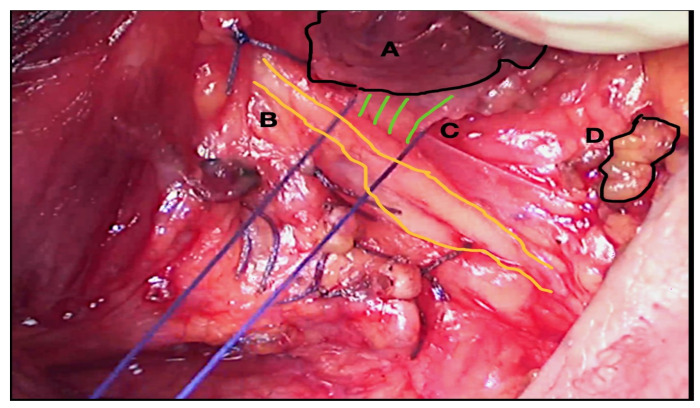
RLN dissection with optical magnification 4.5×. A Thyroid gland. B Right-sided RLN identified and dissected among ligatures. C Berry’s ligament. Excessive traction during ligation may cause RLN injury. D Inferior parathyroid gland.

**Table 1 medicina-58-01560-t001:** The comparison between the OM and IONM group demonstrated statistically relevant data regarding the duration of the operation, transient hypoparathyroidism, and length of hospitalization. Demographics, pathophysiological characteristics, and results in the two groups. Abbreviations: IQR—interquartile range, CM—cervicomediastinal, F—female, FT-UPM—follicular tumor of uncertain malignant potential, *p*-value, ns—non statistically differences.

Demographics and Pathophysiological Characteristics	OM Group (*n* = 50)	IONM Group (*n* = 50)	Total (*n* = 100)	*p*
Gender (F, %F)	41 (82%)	42 (84%)	83 (100%)	ns
Age (median, IQR)	53.5 (42–62)	55 (43–65)	55 (43–65)	ns
Admission diagnosis (n, %) Multinodular goiter Multinodular CM goiter Papillary carcinoma	27 (54%) 21 (42%) 2 (4%)	22 (44%) 25 (50%) 3 (6%)	49 (49%) 46 (46%) 5 (5%)	ns
Hyperthyroidism (n, %)	7 (14%)	6 (12%)	13 (13%)	ns
FNA with cytological examination Absent	19 (38%)	28 (56%)	47 (47%)	ns
Histology Papillary carcinoma Follicular carcinoma Follicular adenoma FT-UPM Hyperplasia	20 (40%) 0 (0%) 9 (18%) 1 (2%) 20 (40%)	10 (20%) 5 (10%) 0 (0%) 0 (0%) 35 (70%)	30 (30%) 5 (5%) 9 (9%) 1 (1%) 55 (55%)	<0.05
Thyroiditis (*n*, %) Absent Chronic lymphocytic Chronic nonspecific Chronic autoimmune	36 (72%) 6 (12%) 6 (12%) 2 (4%)	36 (72%) 8 (16%) 3 (6%) 3 (6%)	72 (72%) 13 (13%) 8 (8%) 7 (7%)	ns
Duration of the operation in minutes (median, IQR)	80 (70–90)	100 (90–119)	90 (75–102)	<0.05
Length of hospitalization in days (median, IQR)	2 (2–2)	4 (4–5)	3 (2–4)	<0.05
Cortisone therapy (*n*, %) Intraoperative Postoperative	50 (100%) 1 (2%)	50 (100%) 8 (16%)	100 (100%) 9 (9%)	<0.05
Hypoparathyroidism (*n*, %) Absent Transient Permanent	33 (66%) 17 (34%) 0 (0%)	44 (88%) 6 (12%) 0 (0%)	77 (77%) 23 (23%) 0 (0%)	<0.05
Dysphonia (transient) (*n*, %) Absent Present	45 (90%) 5 (10%)	44 (88%) 6 (12%)	89 (89%) 11 (11%)	ns
RLN paralysis (temporary) (*n*, %) Absent Monolateral Bilateral	50 (100%) 0 (0%) 0 (0%)	48 (96%) 0 (0%) 2 (4%)	98 (98%) 0 (0%) 2 (2%)	ns

**Table 2 medicina-58-01560-t002:** Results of logistic backward analysis for transient dysphonia and transient hypocalcemia. Abbreviations: OR—odds ratio, IC—confidence interval, *p*-value, Adj OR—adjusted odds ratio.

**Transient** **Dysphonia** **Gender**	**Univariate Analysis**			**Multivariate Analysis**		
	**OR**	**IC (95%)**	** *p* **	**Adj OR**	**IC (95%)**	** *p* **
F M	1 2.53	0.68–9.44	0.28	1 5.19	0.99–27.18	0.05
**Transient** **Hypocalcemia** **Group**	**Univariate Analysis**			**Multivariate Analysis**		
	**OR**	**IC (95%)**	** *p* **	**Adj OR**	**IC (95%)**	** *p* **
IONM OM	1 3.78	1.34–10.62	0.02	1 4.11	1.42–11.93	0.01

## Data Availability

Not applicable.

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
