# Peer review of "Intraoperative Neuromonitoring and Optical Magnification in the Prevention of Recurrent Laryngeal Nerve Injuries during Total Thyroidectomy"

_medicina, 2022, doi:10.3390/medicina58111560_

Round 1

Reviewer 1 Report

 The recurrent laryngeal nerve (RLN) paralysis is a known complication during thyroidectomy. Intraoperative neuromonitoring (IONM) and optical magnification (OM) are the two methods for RLN identification and dissection. The authors evaluate the influence of the two techniques on the incidence of RLN paralysis and determine correlations regarding common outcomes in thyroid surgery.  No statistically significant differences in the presence of hyperfunction, thyroiditis and preoperative cytology were found in between the two methods.  The distribution of the nerves-at-risk (malignancy, multinodular cervicomediastinal goiter, hyperthyroidism) was also similar in the two groups that undergoes OM and IONM.

The results are presented well, and the findings are novel. However, I have a few suggestions to improve the manuscript.

In the abstract section (line 28), the authors mentioned about the less experienced surgeons. Please mention the criteria that the authors have used to describe the experience of the surgeon. You can also remove the term ‘less experienced’ since it is an abstract term.

Similarly, please provide information about the total percentage of IONM and OM done on patients worldwide. This would help the readers to understand how frequent the IONM and OM is used in patients. Add this information in the discussion section.

Author Response

First of all I would like to tank reviewer n.1 for his/her comment that allowed as to improve the manuscript. I will respond point by point to the reviewer's comments.

 Question n. 1: In the abstract section (line 28), the authors mentioned about the less experienced surgeons. Please mention the criteria that the authors have used to describe the experience of the surgeon. You can also remove the term ‘less experienced’ since it is an abstract term.

Reply: The only criteria to describe the experience of the surgeons was meant by the number of total thyroidectomies performed as a first operator. We totally agree to remove that term from the abstract as it does not represent our series.

Question n. 2: Similarly, please provide information about the total percentageof IONM and OM done on patients worldwide. This would helpthe readers to understand how frequent the IONM and OM isused in patients. Add this information in the discussion section.

Reply: Worldwide the use of IONM is widespread, with some statistical differences among countries. For instance, in German more than 90% of thyroidectomies are performed using IONM; in the United States it is used in up to 40% of operations, while in France the rate is nearly 15% of thyroid procedures [6,18].  In Italy, from 2007 to 2013 the number of thyroidectomies performed with IONM steadily increased from 253 to 5100 respectively, reflecting the rapid popularization of this method [18].

On the other hand, magnification techniques (surgical loupes or surgical microscope) are not yet standardized in total thyroidectomy. Only few data are available in the literature, as reported by a recent meta-analysis [17], with different methods and degrees of magnification applied in heterogeneous studies, which only 3 were RCTs. Therefore, more prospective trials are necessary to define better the impact of magnification techniques.

I have added this information in the discussion section.

Reviewer 2 Report

The authors compared IONM and OM in total thyroidectomy to evaluate the efficacy on preserving RLN. I have some queries.

1. Usage of OM in thyroidectomy has been ordinary. So design of the study seemed inadequate, which compared the methods between IONM and OM. I think that IONM plus MO and OM should be compared.

2. Was this study RCT?  If not, how did you assign the patients into the groups?

3. As the reason of high rate of transient hypocalcemia in OM group, the authors described the the high rate of cancer, which required a meticulous dissection. However, I wonder the pathological diagnosis was confirmed after operation and,  if so, the operative methods should be comparable between the groups.

Author Response

First of all I would like to tank reviewer n.2 for his/her comment that allowed as to improve the manuscript. I will respond point-by-point to the reviewer's comments.

 Question n.1: Usage of OM in thyroidectomy has been ordinary. So design of the study seemed inadequate, which compared the methods between IONM and OM. I think that IONM plus MO and OM should be compared.

 Reply: With respect to the reviewer's comment, we fully disagree, because as stated by the literature [ 5, 15, 16, 17] the use of magnification techniques is not yet well defined in total thyroidectomies, while the use of IONM is quite widespread [6,18]. We are aware that more prospective trials are necessary to validate the use of OM in thyroid surgery and a study comparing the results of OM + IONM versus OM alone is about to be started in our institution.

We are happy to provide this information to the discussion section.

Question n.2: Was this study RCT? If not, how did you assign the patients into the groups?

Reply: This is a prospective study where the patients were casually and alternatively assigned to one of the two groups (OM or IONM), based on the progressive booking number to undergo total thyroidectomy in our institution, and up to the number of 50 patients for each group.

This information will be supplied in materials and methods.

Question n.3: As the reason of high rate of transient hypocalcemia in OM group, the authors described the high rate of cancer, which required a meticulous dissection. However, I wonder the pathological diagnosis was confirmed after operation and, if so, the operative methods should be comparable between the groups.

Reply: Regarding the high rate of transient postoperative hypocalcemia in OM group (34% patients vs 12% patients), we assume that this could be explained by the histopathology reports. The histology revealed a statistically significant difference on thyroid cancer in the OM group (40% vs 30%, p<0,05) and this data was consistent with the results of FNA and cytology performed preoperatively on those patients. However, the presence of cancer did not modify the operative method on the OM group as a lymphadenectomy wasn’t required to be done (one of our exclusion criteria of the study). Thus, the finding of nodules suspected of malignancy requires a meticulous dissection to ensure radical excision and the maneuvers performed to preserve the parathyroid glands might compromise temporary the vascularization and, therefore, their function. This is in line with our previous studies [44].

We add this information on the discussion of the results.

Reviewer 3 Report

Historically, RLN paralysis is not a common complication when performed by experienced surgeons.  Therefore, a large number of patients in each group would be required in such a study being described in order to reach a significant difference if one existed.  What was the projected difference in incidence of paralysis between the two arms and how many patients would be needed to reach an acceptable power for the observation?  This does not appear to have been determined in the planning of the trial. Furthermore, there was no indication of whether patients were randomized to each of the treatment arms.  If not, one cannot be certain there were potential inherent biases when patients were selected.  Also, this applies to the selection of surgeon for each treatment arm.  One can only be left with the conclusion that the design of the study precludes the ability to demonstrate a significant difference between the incidence of nerve paralysis when using OM versus OM+IONM. The data produced by the trial is interesting but does not prove anything regarding the superiority of the treatment technique.   

The authors do not provide any information on whether the RLN lost its stimulatory signal upon completion of the resection.  This has particular relevance for the 2 cases in which there was bilateral paralysis.

Author Response

First of all I would like to tank reviewer n.3 for his/her comment that allowed as to improve the manuscript. I will respond point-by-point to the reviewer's comments.

 Question n.1: What was the projected difference in incidence of paralysis between the two arms and how many patients would be needed to reach an acceptable power for the observation?

Reply: The most common complications in thyroid surgery are RLN lesions and hypocalcemia. As reported in discussion “During total thyroidectomy the risk of permanent RLN paralysis, even in experienced centers, is estimated to be nearly 1% for multinodular goiters [8], while in case of reoperations can be quite high (30%) [9]. A recent systematic review reported an incidence of temporary and permanent RLN palsy of 9.8% and 2.3% respectively [10]. Additionally, data from a wide national database documented a twofold increase of RLN paralysis rates when patients routinely undergo postoperative laryngoscopy [3]. Therefore, the incidence of RLN palsy is often underestimated and total thyroidectomy is yet not free of RLN complications.” As stated on the discussion of our report, we recognize some limitations of our study such as “…the use of IONM in intermittent modality only and to the limited number of patients enrolled… Therefore, a new study with the use of IONM in continuous modality and with a larger cohort of patients could certainly provide us further informations.”

In our results we reported that in group IONM 4% of the patients had temporary bilateral RLN paralysis documented postoperatively versus 0% in the OM group. However, no statistically significant difference was found between the two groups (p>0.05). As further explained in the discussion “Despite this unexpected result in our series, we must not forget that the use of IONM is not yet correlated with zero incidence rate of RLN lesions, as evidenced by the literature. For instance, a meta-analysis of 30992 total thyroidectomies reported that the incidence of bilateral RLN paralysis in the group of IONM was average 2.43% versus 5.18% in the group without IONM [26].” This is a statistically significant result in a large cohort of patients, however, not so far from our results regarding the incidence of RLN palsy, that was proven without statistical difference.

Question n.2: there was no indication of whether patients were randomized to each of the treatment arms. If not, one cannot be certain there were potential inherent biases when patients were selected.

Reply: This is a prospective study where the patients were casually and alternatively assigned to one of the two groups (OM or IONM), based on the progressive booking number to undergo total thyroidectomy in our institution, and up to the number of 50 patients for each group. This information has been added in materials and methods at line 95.

Furthermore, the surgery was performed by the same experienced endocrine surgeons, as stated on the methods of our report and the two groups of our study is IONM versus OM alone and not OM+IONM versus OM as assumed by the reviewer. However, we are happy to announce that a study comparing the results of OM + IONM versus OM alone is about to be started in our institution. This information has been added in discussion section at line 320.

The reviewer claims that the data of our study does not prove anything regarding the superiority of the treatment technique. As frequently occurs in the literature, the studies do not have to prove necessarily the superiority of one treatment technique on another. In our series we conclude that (line 336) “Although the 2 cases of temporary bilateral RLN paralysis in the IONM group, not statistically significant difference was found and the overall risk of RLN lesions remains comparable in the two groups. From our experience emerges that both techniques can be considered a standard in the future, as complementary methods, at least in highly specialized centers…”

Question n.3: The authors do not provide any information on whether the RLN lost its stimulatory signal upon completion of the resection. This has particular relevance for the 2 cases in which there was bilateral paralysis.

Regarding the RLN loss of signal in the IONM group, we don’t’ agree that we don’t provide any information and we report exactly what is stated in the discussion of our study: “In both cases of temporary bilateral RLN paralysis, at the end of each thyroid lobe dissection, the RLN function was tested with the IONM and was found preserved, otherwise a total thyroidectomy should have been performed [21]. Only after revised the hemostasis on the trachea axis, we recorded a bilateral LOS of type 1. Hence, we must assume that the RLN injuries were likely caused by the transfer of thermal energy during the electrocoagulation maneuvers, and this might have affected, bilaterally, the intralaryngeal branches of the RLNs. As reported by the literature, thermal stress is one the most significant injuries in terms of risk for permanent VC palsy [7,12, 21] and, in accordance with other authors, moderate to severe thermal injuries of the RLNs involve an increase rate of permanent VCs palsy that needs longer time to recovery [41]. Regarding the other 48 patients of IONM group, we recorded 3 cases of LOS type 2 during the dissection of the second lobe, which however did not result in a postoperative RLN deficit, an event reported also in the literature. Indeed, a LOS registered intraoperatively with intermittent IONM is not associated with vocal cords palsy in up to 33% of the cases [42]. We, therefore, believe that these cases were due to a malpositioning of the surface of the electrodes, which probably occurred by excessive anterior medial traction on the thyroid lobe during dissection of Berry’s ligament [7,13].”

We feel that the  reviewer’s comments improved our manuscript and we hope that the manuscript in now suitable for publication.

Round 2

Reviewer 2 Report

I verified the authors responded the each commment well and understood my inquiry. I think that the article is acceptable.

Author Response

Thank you very much for the review, which has certainly improved the manuscript.

Reviewer 3 Report

I apologize for misinterpreting the difference in technique being applied between the 2 arms of the trial. However, to make this more clear, I would add a statement saying that the the use of optic magnification was not used in the arm that had nerve monitoring.  I agree that subsequent investigations should include OM and IONM in the same arm since this represents the 'state of the art' in most developed countries.

In the methods section, the authors should include a description of how they determined the number of patients required for each arm based on their determination of what percentage difference would be statistically significant. In other words, based on the patient numbers, what was the power of the study? I suggest that they consult a statistician for assistance.  

Author Response

Thank you very much for the review. I will respond point-by-point to the reviewer's comments.

 Question n.1: I would add a statement saying that the use of optic magnification was not used in the arm that had nerve monitoring.

Reply: As suggested, I add the sentence “In the IONM group, the optical magnification was not used” in materials and method section, line 56.

Question n.2: In the methods section, the authors should include a description of how they determined the number of patients required for each arm based on their determination of what percentage difference would be statistically significant. In other words, based on the patient numbers, what was the power of the study?

Reply: Regarding the power of our study, we would like to ensure the reviewer that all data reported were evaluated by experienced statistician from of our institution, who carried out the statistical analysis, reported in materials and methods section and results section and accompanied by tables 1,2,3 and figure 1 to explain better the accuracy of the data. “Statistical analysis was performed using SAS software, version 6.1. Continuous variables were expressed as median (interquartile range, IQR), while categorical variables as proportions and percentages. Student's t test was used to compare continuous variables with normal distribution. The chi-square and Fisher’s exact tests were used to evaluate differences between categorical variables, towards the methodology of regression logistic model and backwards analysis, therefore verify the accuracy of the outcomes and estimate the odds ratio (OR). The level of significance was set at p < 0.05.”

We would like to highlight that, as a good medical practice and as stated by the literature, a prior power analysis should be addressed before every clinical study, with the intend to avoid studies that can't address their primary question or studies that waste precious resources by being larger than they need to be. We agree that statistical power enables you to find the accurate sample size. If the sample size is too small, the results may be invalid. Contrary, if the sample size is too big, you may obtain false positive results. For this reason, it is useful to address statistical power before undertaking any clinical study. In our study and with the valid contribution of our statistician, we were able to calculate the minimum sample size required to detect the effect size of interest, and that was exactly the number of 50 patients for each group, which means 200 RLN at risk, therefore we set the level of significance at p< 0.05.  We were happy to reject the hypothesis 0 at p<0.05, so conventionally we set power at 0.8 to give a 4:1 false negative/false positive risk ratio[(1-0.8)/0.05) ]. This was applied to all algorithmic models, as reported on figure 1 and represented by the ROC curves.

We feel that the  reviewer’s comments improved our manuscript and we hope that the manuscript in now suitable for publication.